# Varying Opinions about Animal Welfare in the Australian Live Export Industry: A Survey

**DOI:** 10.3390/ani10101864

**Published:** 2020-10-13

**Authors:** Patricia A. Fleming, Sarah L. Wickham, Anne L. Barnes, David W. Miller, Teresa Collins

**Affiliations:** Harry Butler Institute and Food Futures Institute, Murdoch University, WA 6150, Australia; sarahlwickham@outlook.com (S.L.W.); a.barnes@murdoch.edu.au (A.L.B.); d.miller@murdoch.edu.au (D.W.M.); T.Collins@murdoch.edu.au (T.C.)

**Keywords:** social licence, physiology, behaviour, quality assurance, welfare indicators, benchmarking

## Abstract

**Simple Summary:**

A social licence for the continuance of the live animal export trade requires confidence in the transparency and validity of measures used to monitor animal welfare and regulate the industry. Objective measures that indicate animal responses to environmental conditions are also required at each stage of the live export chain to enable continuous improvement. As different stakeholder groups are likely to have varied opinions according to their perspective of the industry, understanding these should allow clarity of messages and improved communication. We therefore constructed an open survey of animal welfare assessment methods and reviewed the responses of 921 volunteers. Outcomes demonstrated that there is common interest in developing appropriate animal welfare assessment tools, although there is some misalignment in what was perceived as the ideal format for those tools. This imbalance is due to measures that are “expected” to be included, but which are not always practical or important in the context of live animal export. Through this gained insight, we propose the development of a set of animal welfare assessment tools that have broad acceptance and that are applicable in the industry context. Ongoing work is required to establish the validity, applicability and relevance of these measures.

**Abstract:**

There is significant public interest in the Australian live animal export industry and a need to develop a program that can measure and monitor animal welfare throughout the supply chain. An online survey of stakeholder opinions of this industry and animal welfare measures was carried out in 2015 with 921 respondents: 30% from the public who identified as animal welfare advocates (AWAs); 44% from the public who did not identify as AWAs (general public; GP); 26% live export industry (LEI) workers. AWA and GP respondents expressed greater concern than LEI respondents for animal welfare throughout the supply chain but had less concern for animals at Australian feedlots than in other parts of the supply chain. The majority of AWA and GP respondents believed data collected on animal welfare should be made public and should be collected by independent welfare officers and used to regulate the industry and impose penalties for poor welfare. LEI workers believed that data should be confidential, collected by LEI workers and used by the industry to self-regulate. AWA and GP respondents rated the importance and practicality of a number of welfare indicators greater than LEI workers, while respondents shared an analogous view of the importance and practicality of these indicators. Results can be used to develop welfare assessments that ensure a better understanding between industry members and those not in the industry, while facilitating welfare improvements and promoting greater transparency for the live export industry.

## 1. Introduction

The Australian live animal export industry is a major contributor to the Australian economy, in 2018/9 worth AUD 1.79 billion to the Australian economy [1]. In 2018/9, Australia exported via sea and air a total of 2.32 million head of livestock; this total was composed of 1.26 million cattle worth AUD 1.64 billion, 0.989 million sheep worth AUD 142 million, and 18,650 goats worth AUD 7.2 million [1]. However, the value of the live export industry goes beyond the price of the sales alone, with almost 10,000 people employed in the live export of cattle industry across Australia, including livestock producers, transporters, and exporters [1,2]. Many of these jobs are located in rural and regional areas across the country and, therefore, many communities directly and indirectly benefit from this trade. 

The general public currently seeks greater assurance about the management and welfare outcomes of exported animals [3]. Livestock welfare provokes wide social discussion, particularly when the media deal with incidents of inappropriate conditions or harm to production animals [4]. The rise of animal advocacy groups and the broad recognition of animals as sentient beings have led to the need for a better understanding of the various perceptions of animal welfare that exist among stakeholders [3]. Increased public concern about intensive farming and long-distance transport of animals in Australia is apparent [5]. These trends in public concerns cannot be ignored because they potentially drive ethical consumerism and can influence animal welfare law and regulation [6], as demonstrated by legal regulations that have been enacted in the European Union (EU) for intensive production systems [7] and in Australia in response to the McCarthy Review of live sheep exports [3,8]. 

Calls from some societal groups for the cessation of the live sheep trade from Australia to the Middle East are of significance to the live sheep export industry [9] and there is currently an activist campaign to have the trade banned [5]. Societal perceptions are influenced by multiple factors, including personal convictions, values, norms, knowledge and interests [10]. Thus, identifying issues of high- and low-potential conflict regarding the live export industry is an important first step to building public confidence in the industry and modifying livestock production practices that cause high concern. A Eurobarometer study published in 2005 (covering 31 European countries) indicated that 82.3% of Europeans rated farm animal welfare within the range of moderate to very bad. Additionally, 89.3% indicated they did not receive enough information concerning the welfare conditions of animals farmed in the EU, illustrating the need for more and clearer information [11]. On the contrary, producers have shown a more positive perception of farm animal welfare, and studies have explained that the opposite perceptions between farmers and non-farmers can be explained by a different interpretation of welfare
[10,12]. In a 2018 Australian survey of 1531 respondents commissioned by the Commonwealth government, 95% of people viewed farm animal welfare to be of concern and 91% wanted at least some reform to address this [3]. In addition, 66% of respondents stated that live exports should be banned [3]. Therefore, also surveying the perceptions of animal welfare from respondents within the Australian live export industry in terms of their underlying values and preferences for welfare indicators would be informative. 

In Australia, animal welfare is important to the live export industry, in terms of both economic return and community attitudes. Meeting the demands of industry, consumers, and the broader public is an ongoing challenge for livestock industries [13,14]. Societal demands for sustainable and ethical animal production systems and practices will continue, and therefore the industry must remain proactive in their effort to ensure the welfare of the animals [9]. Additionally, understanding the values and beliefs of industry, consumers and the broader public are critical in addressing challenges and implementing change [4]. Ideally, the concept of a welfare assessment program based on a broad set of indicators that can operate as an industry-driven management system is a logical first step that can promote continuous improvement over and above legislative requirements. Identifying the social acceptance of different animal welfare measures, in terms of practicality and relevance, is therefore the first step towards developing such a benchmarking system.

A number of surveys and reviews have examined public perception in terms of belief on whether the live export trade should persist or not [3,15,16]. However, the attitudes toward animal welfare and the perceptions towards its assessment within the industry have not been investigated. Frameworks for animal welfare assessment on Australian farms [17] and in export supply chains [18] are under development and, once adopted, should bring more understanding about the affective state of animals in production systems, and potentially alter public perception. To the authors knowledge, despite the public concern, there is little transparency about the daily routine care of animals on board livestock vessels. 

Public perception is likely to be strongly influenced through the media and public interest in the welfare of the animals rather than knowledge of current practice per se, and therefore a comparison between animal welfare advocates, the general public, and workers within the live export industry can prove useful in terms of identifying where similarities and differences in opinion lie. Understanding the perceptions of people from different countries or cultures will facilitate the development of welfare standards [6,7], especially for an industry whose stakeholders span multiple cultures and belief systems. We surveyed stakeholders from multiple points of the export chain to determine their perceptions of animal welfare in order to contribute towards identifying specific animal welfare measures that are perceived as both important and practical. 

## 2. Methods

A survey instrument (Appendix A) was distributed to key stakeholder representatives, including members of the general public, animal welfare groups and industry workers. The survey was constructed using SurveyGizmo (www.surveygizmo.com) using a combination of multiple choice and visual analogue scale (VAS) questions, with options to add comments where appropriate. 

The survey was advertised on social media, local and rural newspaper media releases, industry newsletters, and via personal contacts. All voluntary respondents that completed the survey between March and June 2015 were accepted. Respondents were asked whether they worked in the live export industry or not. Respondents who did not work in the live export industry were then asked what their interest in the industry was (Q19); one of the choices was whether they identified themselves as animal welfare advocates. These questions were used to categorise all respondents into three categories: respondents who currently worked in the live export industry (LEI), members of the public that identified themselves as animal welfare advocates (AWAs), or members of the public that did not identify themselves as animal welfare advocates (hereafter “general public”; GP). 

The survey collected information on respondents’ demographics (age, gender, location) and experience with the live export industry. They were asked to rate how important (using a VAS—not important to important) animals are for food, clothing, companionship, working, entertainment, performance, and as a religious symbol. Non-LEI respondents were asked to select from options to indicate reasons for their interest in the live export industry (where they could select more than one option). 

All respondents were asked where they sourced their information on the live export industry, with more than one option selectable, and respondents were asked a number of questions regarding their opinion of animal welfare within the industry. They were also asked to indicate whether they were always, frequently, sometimes or never concerned about the welfare of cattle and sheep for seven locations: at a feedlot in Australia; during loading or unloading at the port (sea); on-board ship during a voyage; on-board aeroplane during a flight; during truck transport overseas; at an overseas feedlot; during slaughter overseas.

Respondents were asked to rate the importance (using a VAS) of time that animals spent at seven points in the live export process: during truck transport, onboard a truck waiting to unload, onboard a ship docked at port, or on voyage, onboard an aircraft during a flight, and at feedlots in Australia and overseas.

Each respondent was asked to identify whether or not (yes or no) they thought a series of 35 potential animal welfare measures were (i) important or (ii) practical to measure. They then used a VAS to identify the importance of another 26 specific measures for the welfare of livestock in the live export industry.

We also asked respondents to indicate who they believed should be assessing animal welfare as part of the live export industry and how they wanted to see that information used: respondents were asked to select whether they thought that data on the industry should be made publicly available or should be confidential to the industry and government only. 

Descriptive statistics were calculated for demographic answers. Comparisons between the three stakeholder groups (AWA, GP and LEI) were carried out using χ2 analyses with expected values calculated assuming that there were no differences between the stakeholder groups (i.e., responses were distributed proportionally to the total number of respondents for each stakeholder category). Responses on a VAS were converted to numerical data where the minimum = 0 and maximum = 100; these data were compared by Kruskal–Wallis ANOVA to compare between stakeholder groups.

## 3. Results

### 3.1. Demographics

Almost a third of the 921 respondents identified as animal welfare advocates (AWA; n = 279) and were distinguished from the remaining general public (GP; n = 401). This division resulted in a reasonably even split between the three stakeholder categories (30% AWA, 44% GP, 26% LEI). LEI respondents (n = 241) included exporters, inspectors, researchers, producers, transporters and abattoir workers (Figure 1). 

The majority of the AWAs (90%) indicated that they were interested in the live export industry due to their concern for animals compared with three quarters (77%) of the GP respondents (*χ^2^*_1_ = 19.47; *p* < 0.001; note that LEI workers were not asked this question). Other options selected (proportions not different between AWA and GP stakeholder groups) were that they had a friend/relative working in the live export industry (11%), were connected with the industry through research (7%) or involved in journalism (3%). While we did not ask respondents to specifically identify whether they were producers or not, 42% of the 107 respondents who selected “other” wrote comments that identified them as producers. Example comments in the “other” selection include: 


*“Lived in rural areas for many years with first-hand knowledge of animal transport practices and standards.”*
AWA respondent


*“Work in the livestock industry.”*
AWA respondent


*“As a compassionate human being I care about all animals. It is a mark of our humanity that we treat all living things with care and respect.”*
AWA respondent


*“Citizen of country that exports livestock - worried about economic benefits of this”*
GP respondent


*“Our family farm does not sell cattle direct to live export but what is happening in the export trades directly impacts on all Australian farmers”*
GP respondent


*“I think live export is important, but represents very low value product and not a very good long term outlook.”*
GP respondent


*“I work for a company that receives live export and it’s an important part of the Australian Beef industry*
GP respondent


*“Just being a normal person who understands the necessity of animals to our whole existence”*
GP respondent


*“I want to see the LE industry stop for good and as quickly as possible. There are other options - move to a chilled trade only, I believe it is only greed and laziness that is preventing this from happening.”*
GP respondent


*“I am a sheep farmer, even though I do not supply to Live Ex (I’m in the South) I believe that the live export industry is an important link in the chain that is Aus Ag.”*
GP respondent


*“Concern for animals and concern for Australian economy and the lost opportunity for value-adding through local slaughter”*
GP respondent


*“The live export trade represents one of our society’s most egregious forms of animal use.”*
GP respondent

The majority of respondents were both born and worked in Australia, with approximately 23% born outside of Australia. There were differences in where respondents lived (χ22 = 92.05; *p* < 0.001), with more LEI respondents living in rural areas and more AWA respondents living in urban areas than predicted from a random distribution amongst the categories (rural percentage: AWA 23%, GP 40%, LEI 65%). 

Respondents ranged from 18 to 71+ years of age (Figure 2). Two thirds (68%) of respondents were female and one third (32%) male. The sex ratio was significantly different between the categories (χ22 = 129.16; *p* < 0.001; Figure 2), with more female respondents in the AWA group and more males in the LEI group than predicted from a random distribution amongst the categories. There was no significant difference in the education level of the three stakeholder categories (χ22 = 5.87, *p* = 0.209), with a similar proportion of respondents who had not finished high school (5%), graduated high school (18%), a diploma or degree (54%) or a postgraduate degree (e.g., masters, PhD; 22%). 

### 3.2. Sources of Knowledge Regarding the Live Export Industry

There were significant differences in responses according to where respondents sourced their information on the live export industry. LEI workers were more likely to identify personal experience (91% of respondents) compared with AWAs (59%) and the GP (53%) (χ22 = 103.46; *p* < 0.001). Journalism media (e.g., print, television and the internet) were identified as a source of information by 90% of AWA and 78% of GP respondents, but only 54% of LEI respondents (χ22 = 91.88; *p* < 0.001). Social media (e.g., Facebook, Twitter) was identified as a source of information by 75% of AWA and 51% of GP respondents, but only 27% of LEI respondents (χ22 = 121.29; *p* < 0.001). AWA and GP respondents were also more likely to identify communication with friends (54% and 44%, respectively) compared with LEI workers (39%) (χ22 = 12.50; *p* = 0.002). A third of respondents (36%) identified “scientific journals” (note that we did not prescribe whether this meant popular scientific journals or journals with impact factor and peer-review procedures) as a source of information about the industry, with reasonably similar responses from AWA (42%), GP (34%) and LEI (31%) stakeholder groups (χ22 = 8.47; *p* = 0.015). 

### 3.3. Experience

Surveyed LEI workers spent 63% of their time working with cattle, 32% working with sheep, and 5% with goats. Not surprisingly, although most respondents had seen livestock being transported by truck (94% average), more surveyed LEI workers had visited or seen other sections of the live export industry than AWA or GP survey respondents (Figure 3). This included visiting feedlots (AWA 44%, GP 56%, LEI 84%; χ22 = 89.82; *p* < 0.001), an abattoir (AWA 48%, GP 56%, LEI 82%; χ22 = 68.03; *p* < 0.001), a live export ship (AWA 53%, GP 49%, LEI 63%; χ22 = 11.93; *p* = 0.003), or an aircraft carrying livestock (AWA 6%, GP 6%, LEI 24%; χ22 = 56.73; *p* < 0.001). Only 4 % of AWA, 4% of GP and 8% of LEI workers had never seen or visited any aspect of the industry (χ22 = 6.33; *p* = 0.042); these LEI respondents were employed in administration and management.

### 3.4. Opinions

There were significant differences in rankings between the three stakeholder groups when rating the importance of different animal roles in society (Figure 4). For AWA and GP groups, the role of animals as companions was the most important to them. For all other roles, LEI workers gave higher importance ratings than both AWA and GP groups (*p* < 0.001 for roles other than religious symbol which was *p* = 0.013).

AWA respondents rated (using a VAS) the amount of time that animals spent in any part of the export chain as a more important welfare concern than the other stakeholder groups (Figure 5), although the time spent in Australian feedlots was rated as less important than time spent across the other parts of the chain (Figure 5f). 

The surveyed public (both AWA and GP groups) consistently rated (using a VAS) 26 factors that could influence animal welfare as “more important” than surveyed LEI workers (Kruskall–Wallis ANOVA; *p* < 0.001) (Figure 6). Respondents rated effective slaughter procedures and the attitude and competency of stockpersons as the most important of these 26 factors (Figure 6a–c), and the use of an electric prods and working dogs as the least important (Figure 6y,z), although the scoring of these last two options appeared to be ambiguous (see discussion).

There were significant differences in responses to the question whether stakeholders had concerns about livestock within the live export industry (χ2 analyses for each species/location category, *p* < 0.001; Figure 7). AWAs were largely always concerned (AWA: Always—75% of responses, Frequently—10%, Sometimes—7%, Never—8%). Although Always was also the most common response for GP respondents, just over a quarter of these respondents also indicated that they were never concerned (Always—46% of responses, Frequently—10%, Sometimes—17%, Never—27%). The majority of LEI respondents stated that they never or sometimes had concerns for the welfare of the animals (Always—7% of responses, Frequently—5%, Sometimes—30%, Never—58%) (Figure 7).

The majority of respondents believed that data collected on animal welfare in the live export industry should be made publicly available (AWA 85%, GP 78%, LEI 51%, *p* < 0.001), while LEI respondents more often selected the option that information should be confidential to the industry and government only (AWA 2%, GP 9%, LEI 36%, *p* < 0.001).

Survey respondents were asked how they would prefer to see data collected on animal welfare used in the industry (Figure 8a). The most common reason selected by respondents was to impose penalties and restrictions for poor welfare (selected by 77% of respondents with slight differences between stakeholder groups: AWA 86%, GP 80%, LEI 62%, *p* = 0.013). The three stakeholder groups broadly agreed with the use of animal welfare measures to highlight areas for investment in welfare improvement (no significant difference in proportion of responses *p* = 0.767), which was selected by 71% of respondents. There were also no significant differences in the proportion of respondents who selected to award incentives for good welfare (*p* = 0.291; selected by 60% of respondents) and to highlight areas for research (*p* = 0.120; selected by 53% of respondents). All other options were selected disproportionally by the stakeholder groups. Notably, LEI respondents were more likely to select the option for the industry to self-regulate (AWA 13%, GP 36%, LEI 73%, *p* < 0.001) and less likely to select the options for the government to regulate the industry (AWA 70%, GP 59%, LEI 35%, *p* < 0.001) or provide measures of performance to the general public (AWA 72%, GP 63%, LEI 52%, *p* < 0.039) than AWA or GP respondents. 

Respondents were asked who they would prefer to see collecting data on animal welfare in the industry (Figure 8b). The most common option selected by respondents was independent welfare inspectors (selected by 67% of respondents with strong differences between stakeholder groups: AWA 79%, GP 68%, LEI 51%, *p* < 0.001). LEI respondents would prefer to see those working within the industry responsible for monitoring animal welfare and were more likely to select stockpersons (AWA 29%, GP 43%, LEI 77%, *p* < 0.001), veterinarians that work for the live export industry (AWA 27%, GP 40%, LEI 55%, *p* = 0.025), and exporters (AWA 20%, GP 32%, LEI 62%, *p* < 0.001). AWA and GP respondents would prefer to see animal welfare monitored by people who are independent from the industry, such as animal welfare inspectors (AWA 73%, GP 62%, LEI 46%, *p* < 0.001) and veterinarians that work for the Australian government (AWA 56%, GP 59%, LEI 55%, *p* < 0.001). The proportions of respondents selecting abattoir animal welfare officers (47% of responses, *p* = 0.291) and veterinarians that work for an overseas government (46% of responses, *p* = 0.491) did not differ between stakeholder groups. Both AWA and GP stakeholders expressed their opinion (via comments) that inspectors need to be impartial from the industry and government, and not penalised for reporting negative welfare, while those working within the industry do not want animal rights/activist groups monitoring welfare. For comments under the Other option, more than 17% of all survey respondents think that it is everyone’s responsibility to monitor and measure animal welfare in the live export industry.

### 3.5. Welfare Measures

The majority of survey respondents generally agreed on the importance and practicality of a number of physiological-, health- and environmental-based welfare indicators (Table 1), although more of the AWA and GP respondents indicated that the measures described were important and practical, compared with LEI workers (χ21 analyses; *p* < 0.05 for each) and more LEI workers rated meat quality as important (*p* > 0.05). 

A greater number of respondents considered injury/wounds and ventilation as important, while fewer considered sneezing and smell as important. Similarly, for practicality, more respondents rated injury/wounds, inability to stand, ventilation and amount of shade/shelter as practical to measure, and fewer respondents identified pain and smell as practical measures. While the majority of respondents agreed that disease and death were important as well as practical physiological welfare indicators, they differed in their opinions for other physiological indicators—fewer LEI respondents identified invasive measures (e.g., heart rate and stress hormones) as important and practical than AWA and GP respondents (Table 1).

## 4. Discussion

As there was a good response rate to our voluntary survey, the opinions of the respondents should be representative of the animal welfare advocates, the general public, and live export industry workers in 2015. Despite differences in demographics, attitudes towards the importance of animal roles in society, and the experiences of each group of survey participants, there was common interest in developing appropriate animal welfare assessment tools for the livestock export industry (LEI). The three groups displayed similarities in perception and expectations of community and industry stakeholders in terms of animal welfare assessment. There was some misalignment in what respondents idealise as the format of those tools. However, much of this misalignment is likely due to the perception that measures that are “expected” to be included are not always practical or important in the context of the live export industry. For example, body temperature and heart rate were seen as only moderately important and not very practical by the LEI group, but much more highly rated by the AWA and GP groups. Importantly, how respondents ranked the practicality and importance of different welfare measures reveals potential future direction for the live export industry, addressing community concerns and allowing advancements in animal welfare. 

### 4.1. Some Expected, and Some Unexpected, Differences of Opinion

Our survey did not set out to determine whether people want the live animal export industry to continue or not—such has already been completed [19,20]. Rather, our survey set out to identify what sort of animal welfare assessment is sought by the general public, animal welfare advocates, and workers within the industry itself. Importantly, there is a need to identify how these different stakeholder groups perceive the importance and practicality of animal welfare assessment methods. 

We identified some differences between the stakeholder groups in their opinion on welfare indicators. Overall, there was good agreement between these groups in how health and environmental welfare indicators ranked in terms of their perceived importance and practicality. Differences in the ranking order of physiological measures could be due to differences in experience within the industry and with experience of collecting these measures themselves. For example, invasive procedures are involved to measure heart rate, stress hormones, and body temperature, and LEI workers may not see the value in collecting them if they are labour intensive and have a negative welfare impact to the animal during collection (i.e., handling, isolation). AWA and GP respondents, on the other hand, may not be aware of the limitations in measuring some of these indicators. Upon further conversation with some respondents, they thought that technology should be used to remotely measure body temperature and heart rate and did not see any limitations in the practicality of measuring them.

We found unexpected responses to the role of electric prods or working dogs for moving animals. Many AWA and GP respondents commented that they believed neither an electric prod nor a working dog should be used, but then rated the use of these tools (using a VAS) as Not important to the welfare of livestock in the live export industry. Therefore, those who rated them as Not important and those who rated them as Important may arguably be conveying the same opinion regarding the welfare implications of their use. Responses for the use of an electric prod and working dog are therefore likely to be an unreliable indication of the welfare opinion of respondents.

There were differences in how AWA and GP respondents sought welfare data to be collected and used compared to LEI workers, with AWA and GP respondents wanting more access to data, the use of independent welfare assessors for data collection, and data to be used as a means to impose penalties for poor welfare. Of interest, 62% of the surveyed LEI workers also supported the idea that the data should be used to impose penalties. This is noteworthy and illustrates desire from within the industry to have improved welfare outcomes and stricter enforcement of regulations. Thus, the industry should be encouraged to use data in new ways, to allow feedback to be received by operators, and to increase public transparency. 

### 4.2. The Importance of Animal Welfare Measures Other than Just Recording Mortality

In 2004, the Livestock Export Program [21] identified seven key indicators of welfare on-board ships based on the opinions from nine stakeholder groups. These were mortality, clinical disease, respiration rate, wet bulb temperature, space allowance, change in body weight, and ammonia levels. The current survey identified differences in our responses compared with the 2004 review, identifying that respondents generally agreed on both the practicality and importance of animal welfare assessment measures; injury/wounds, ability to stand, disease and ventilation were identified as important by the greatest number of respondents for all stakeholder groups. While mortality was the prime indicator in live export identified by Pines et al. [21], for our survey, “death” was ranked 5–7 in the number of respondents indicating its importance—possibly because mortality measures alone are not informative for improving animals’ immediate experience of their environment. Pines et al. [21] identified respiration rate and air temperature as key welfare indicators, which could reflect social awareness of heat stress as part of the live export industry around the time of their survey [22], but these were ranked amongst the bottom half of the measures in our survey. 

Of the top seven indicators from Pines et al. [21], three were resource-based measures compared to our study, which identified only one resource-based measure (ventilation) in the top seven ranked important measures. This may demonstrate a convergence of views between stakeholder groups over time about the relative importance of animal-based measures. A similar conclusion was reached by the European Food Safety Authority released a document in 2012 stating *“animal-based measures are the most appropriate indicators of animal welfare”* because *“animal-based measures aim to measure the actual welfare of the animal and thus include the effects of different input factors”* [23]. 

A farm animal welfare survey by Te Velde et al. [10] indicated farmers and citizens showed similar values with regard to measures of physical health (adequate food and water, heating) but differed in how much they valued behaviour; e.g., citizens included additional values of freedom to move and freedom to fulfil natural desires to which the farmers showed indifference. In our study, “behaviour” was ranked strongly in terms of the number of AWA, GP and LEI workers who identified the measure as *Important*, suggesting all stakeholder groups believe measuring behaviour to be essential. However, caution is needed in drawing conclusions across studies because various stakeholder groups may or may not have similar motives or understanding of the utility of these measures, despite common interests; e.g., farmers and LEI workers may be more economically driven and consumers may be more interested in the supply of healthy inexpensive food, while the general public may be more interested in ethical production systems [12].

A number of additional measures can be considered for further investigation, including new tools that are needed to supplement those currently already carried out by the LEI [18]. We did not specifically ask respondents about the importance of space allowance or ammonium levels, although we asked about a range of similar physical conditions (amount of shade, shelter from weather, amount of light, or air quality and humidity). Providing basic information to the public on space allowance, provision of adequate feed and other environmental provisions, such as air temperature and quality, could be considered. 

### 4.3. How the Live Export Industry has Already Changed

This study captures the views and opinions of animal welfare advocates, the general public and workers associated with the live export industry recorded in 2015. At the time, social media was a major source of information opposing the industry. In the intervening period, there has likely been some changes to stakeholder perception due to media stories threatening the industry’s social license to operate and subsequent changes to government policy. In addition, the Livestock Collective (https://thelivestockcollective.com.au/) media platform was developed, which has 6000+ followers on Facebook and 1700+ on Twitter and this additional source of information about livestock production may have broadened stakeholder views. Media will still likely have a strong influence on public opinion; concern tends to be higher for animals and issues that have received media coverage [3], but the wide impact of these is often short-lived. For example, the Awassi Express voyage to the Middle East in August 2017 resulted in about 2400 sheep mortalities and ignited political and public backlash after video footage highlighting poor animal welfare issues was broadcast on television in 2018 [19]. Such footage is a powerful generation of a negative industry image. Sinclair et al. [19] surveyed public opinion immediately before and after the 2018 Awassi Express exposé [24] and found that an increased proportion of respondents believed the trade should end after this exposé aired. This exposé has contributed to the recognition of a greater need for transparent animal-based welfare measures to be used by industry, especially around thermal challenge as highlighted by the McCarthy Review [8]. 

In our 2015 survey, respondents strongly selected that they would prefer to see independent welfare inspectors collecting data on animal welfare across the live export chain. In 2018, a review of the Australian Standards for the Export of Livestock (ASEL) was completed and independent observer program was introduced, requiring an observer on each shipment to improve regulatory capability and public trust in the industry [25]. In September 2019, *the Live* Animal *Exports Bill 2019* was passed by the Australian Federal Government to introduce independent oversight of the live export industry through the appointment of an Inspector-General [26]. A sustained interest in concern from the general community for animal welfare in livestock production is anticipated [9,17,27]. This growing concern can have an impact on the industry’s social licence to operate. The lack of standardisation of welfare certification schemes involved in international trade worldwide adds to the complexity of providing assurance to the community [28]. The live export industry has demonstrated the ability to respond rapidly to welfare challenges in the past (e.g., [8,22]), and looks to public perception of the trade to guide future management options [29]. Further surveys, such as the recent one carried out in 2020 [29], may be useful to determine a communication strategy, including what specific information the public would prefer to see from the industry, how to educate the public on live export industry practices, and how to demonstrate best practice. It also helps to build understanding of how to meet community and industry expectations in designing animal welfare tools.

## 5. Conclusion

Animal welfare is a high priority issue for the community and there is a growing awareness that changing community values need to be reflected in management practices. A live export industry strategy of continued improvement and increased transparency is important to unite, not divide, the various stakeholders involved. Phillips [15] surveyed views of farmers and other industry personnel, overseas consumers, the Australian public, and veterinarians. He concluded that the debate as to whether Australia should continue with the trade (see also [29]) will be best served by consideration of the interests of all parties, including those of the consumers and animals, which are among the most affected by the trade. Different people will place different weighting on the rights and interests of these groups, but extreme positions are often adopted because people fail to understand or have sufficient concern for the position of all stakeholders in the trade. 

Although the interpretation of a complex and multi-dimensional concept such as animal welfare can be challenging [18], this survey showed little discordance between industry and public perceptions in terms of the types of measures to collect for welfare assessment. However, there was some variance in the views on how such data should be used, and hence there is scope for the industry to bring about changes that will build community trust. Consistently meeting the needs of consumers and the broader public, while maintaining a sustainable industry, will be an ongoing challenge. Hence there is urgent need for transparent and open management that provides good animal welfare outcomes and demonstrates aligned values around the care of animals.

## Figures and Tables

**Figure 1 animals-10-01864-f001:**
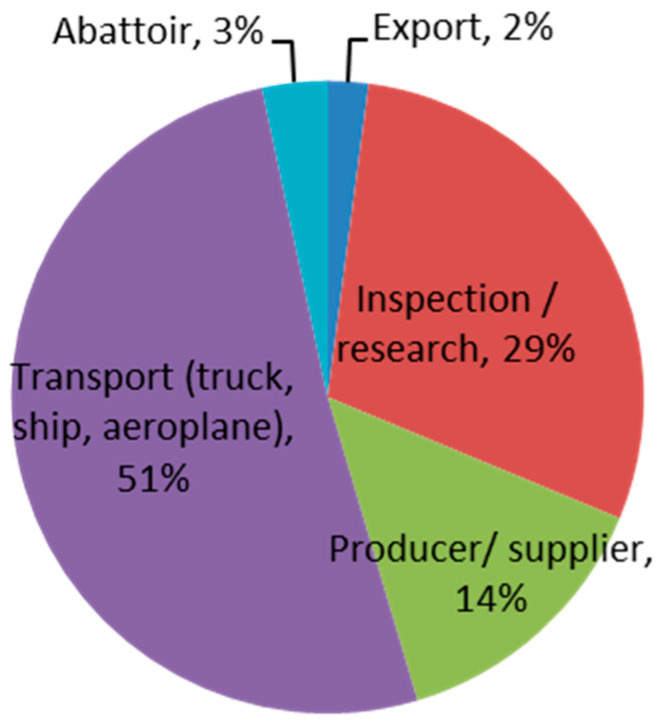
Breakdown of the background of live export industry (LEI) workers who responded to this survey.

**Figure 2 animals-10-01864-f002:**
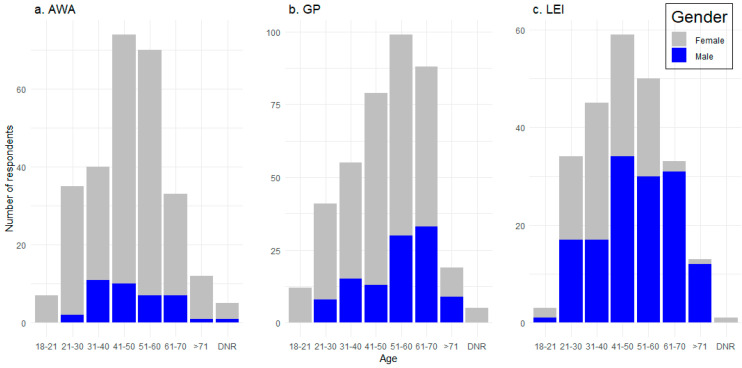
Age and gender of respondents classified as (**a**) animal welfare advocates (AWAs), (**b**) the general public (GP) or (**c**) live export industry (LEI) workers.

**Figure 3 animals-10-01864-f003:**
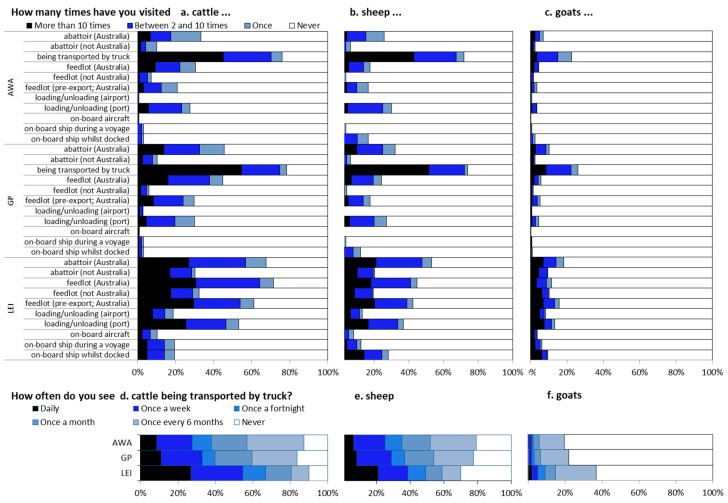
Experiential background of survey respondents grouped into three stakeholder categories: animal welfare advocates (AWAs), general public (GP) or live export industry (LEI) workers.

**Figure 4 animals-10-01864-f004:**
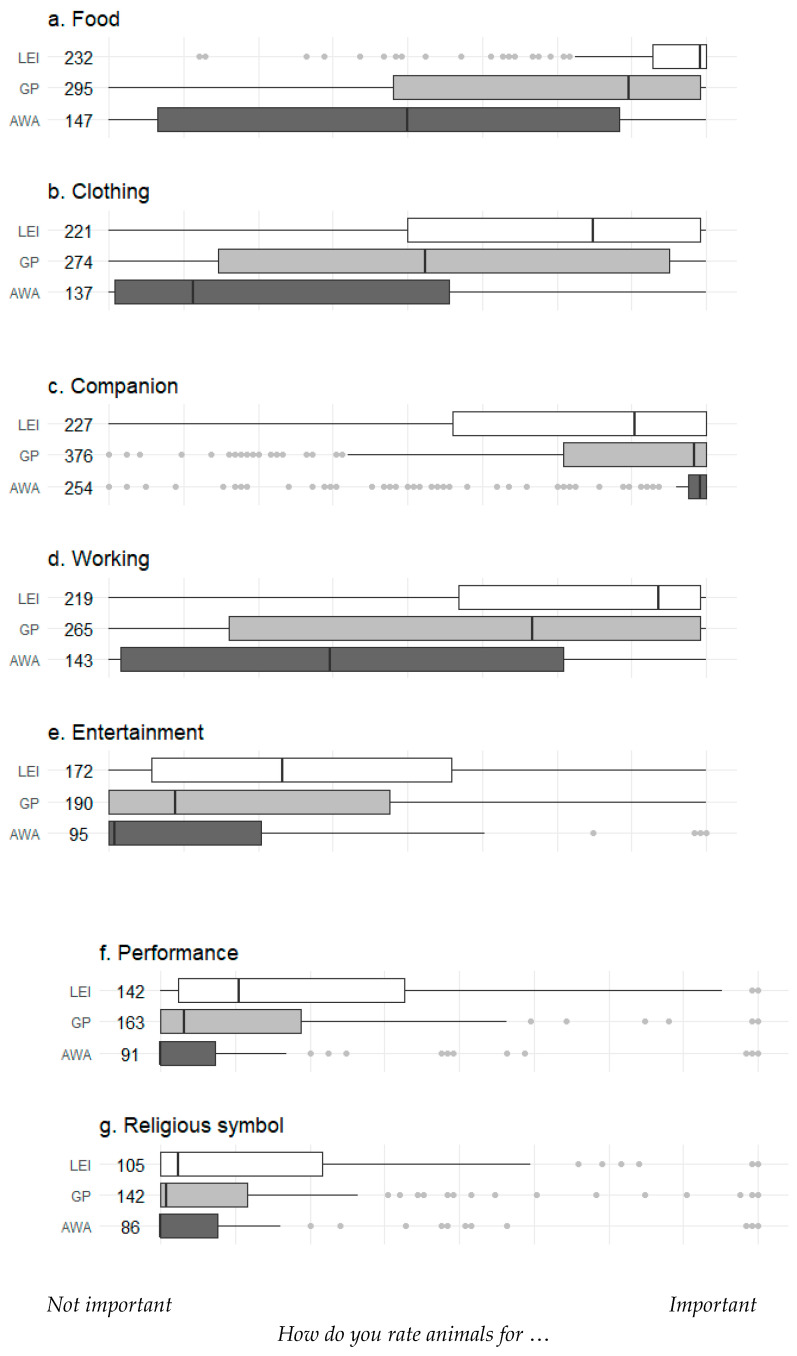
Rating of the importance of animals in various roles by survey respondents grouped into three stakeholder categories: animal welfare advocates (AWAs), general public (GP) or live export industry (LEI) workers. The number of respondents varied for these questions: numbers next to stakeholder groups indicate the number of respondents for each. Data were collected using a visual analogue scale: values are medians (vertical line) and quartiles (boxes) ± non-outlier range (whiskers); dots represent outliers (1.5 × the inter-quartile range). For all these questions, responses differed significantly between the three stakeholder groups (Kruskall–Wallis ANOVA; *p* < 0.001 for a–f and *p* = 0.013 for g).

**Figure 5 animals-10-01864-f005:**
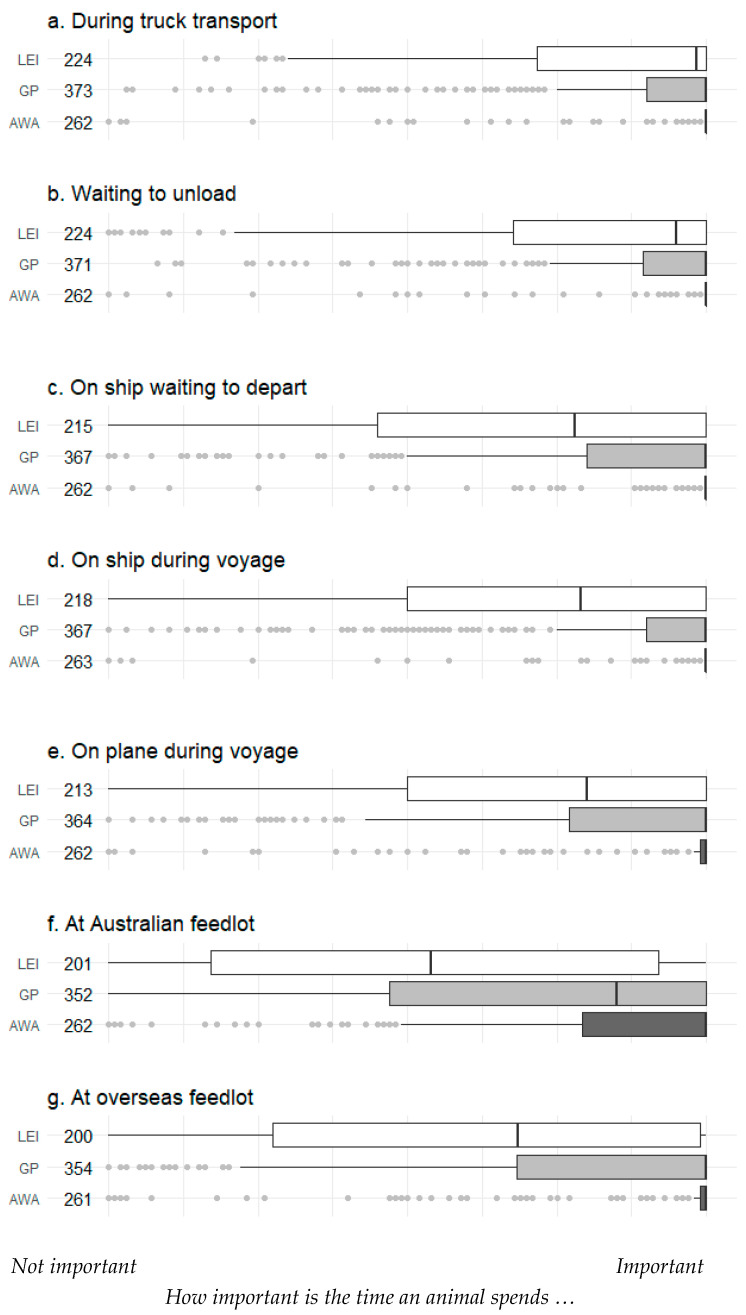
Rating of the importance of how much time an animal spends at each location by survey respondents grouped into three stakeholder categories: animal welfare advocates (AWAs), general public (GP) or live export industry (LEI) workers. Numbers next to stakeholder groups indicate the number of respondents for each question. Data were collected using a visual analogue scale: values are medians (vertical line) and quartiles (boxes) ± non-outlier range (whiskers); dots represent outliers (1.5 × the inter-quartile range). For all these questions, responses differed significantly between the three stakeholder groups (Kruskal–Wallis ANOVA; *p* < 0.001).

**Figure 6 animals-10-01864-f006:**
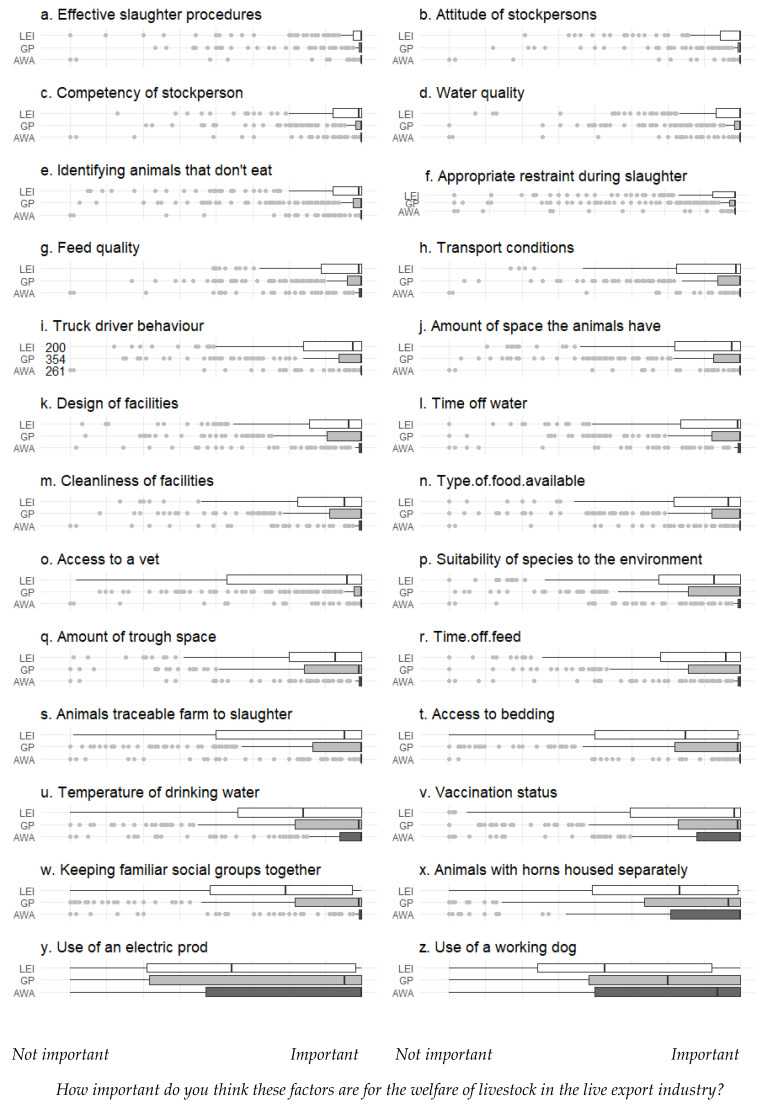
Ratings of the importance of different welfare factors are in the live export industry by survey respondents grouped into three stakeholder categories: animal welfare advocates (AWAs), general public (GP) or live export industry (LEI) workers. Data were collected using a visual analogue scale: values are medians (vertical line) and quartiles (boxes) ± non-outlier range (whiskers); dots represent outliers (1.5 x the inter-quartile range). For all these questions, responses differed significantly between the three stakeholder groups (Kruskal–Wallis ANOVA; *p* < 0.001).

**Figure 7 animals-10-01864-f007:**
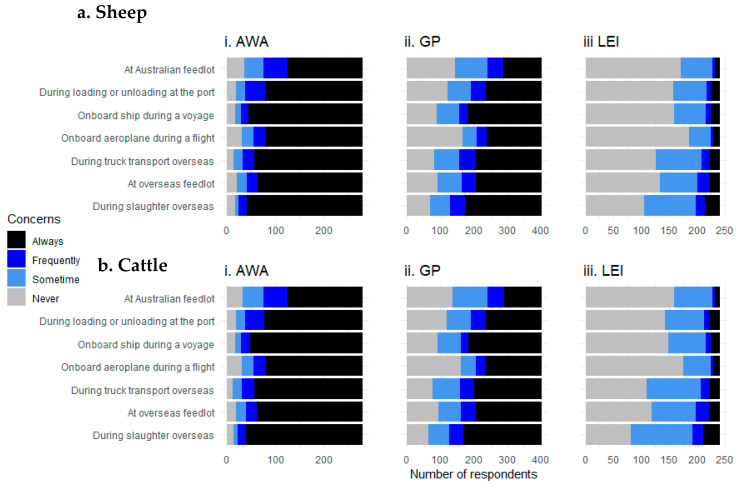
Respondents that expressed concerns for (**a**) sheep and (**b**) cattle among different locations of the live export industry for survey respondents grouped into three stakeholder categories: animal welfare advocates (i, AWAs), general public (ii, GP) or live export industry (iii, LEI) workers.

**Figure 8 animals-10-01864-f008:**
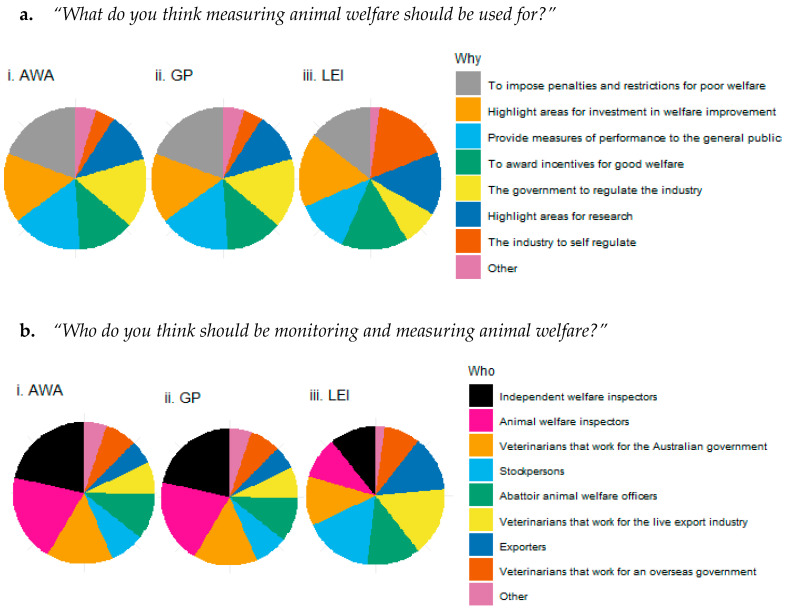
Opinions on (**a**) the purpose of animal welfare assessments and (**b**) who they believe should be responsible for these assessments, for survey respondents grouped into three stakeholder categories: animal welfare advocates (i, AWAs), general public (ii, GP) or live export industry (iii, LEI) workers. Monitoring options are sorted anticlockwise from the most favoured option to the least favoured (across all respondents).

**Table 1 animals-10-01864-t001:** Percentages of each of three stakeholder categories (animal welfare advocates (AWAs), general public (GP) or live export industry (LEI) workers) that indicated they believe 35 animal welfare measures are important or practical. Where there were statistically significant differences in responses between stakeholder groups, the letters (a, b) link percentage values that were not significantly different from each other. Within each of the three question categories, the measures are ordered in decreasing number of respondents who identified them as important (green to red shading for importance, and blue to red shading for practicality).

Question	*“Measure is Important”*	*“Measure is Practical”*
AWA	GP	LEI	AWA	GP	LEI
**Physiological measures**
Parasites	99%		97%		94%		95%		91%		88%	
Physiological status	99%		94%		94%		96%		91%		91%	
Body Condition Score	97%		95%		94%		96%		94%		94%	
Behaviour	99%		96%		91%		95%		90%		82%	
Body weight	92%		90%		81%		92%		90%		86%	
Respiration rate	96%	a	89%	ab	76%	b	88%	a	76%	a	62%	b
Body temperature	95%	a	86%	a	60%	b	87%	a	72%	b	43%	c
Stress hormones	96%	a	83%	a	55%	b	76%	a	55%	b	29%	c
Rumination	91%	a	79%	b	60%	b	78%	a	62%	b	41%	c
Heart rate	96%	a	80%	b	44%	c	79%	a	63%	b	28%	c
Wool length	68%		68%		71%		83%		78%		79%	
Meat quality	49%	b	66%	a	72%	a	62%		64%		67%	
**Health measures**
Injury/wounds	100%		100%		99%		98%		98%		96%	
Presence of disease	99%		100%		99%		97%		96%		94%	
Disease	99%		99%		99%		98%		97%		95%	
Inability to stand	100%		100%		97%		99%		99%		95%	
Infection	99%		99%		97%		96%		95%		89%	
Lameness	100%		98%		97%		98%		97%		94%	
Death	99%		98%		97%		98%		97%		95%	
Pain	99%		94%		84%		87%	a	73%	b	53%	c
Nasal discharge	97%		93%		84%		97%		88%		81%	
Faeces structure	97%		89%		86%		90%		83%		82%	
Coughing	97%	a	91%	a	75%	b	95%	a	82%	ab	73%	b
Vocalisations	97%	a	87%	a	65%	b	93%	a	81%	a	65%	b
Sneezing	93%	a	78%	b	53%	c	93%	a	74%	b	56%	c
**Environmental measures**
Amount of ventilation	100%		99%		100%		98%		97%		97%	
Air quality	100%		97%		94%		96%		93%		85%	
Amount of shade	100%		97%		92%		98%		98%		90%	
Amount of shelter from weather	99%		97%		92%		98%		96%		90%	
Air temperature	99%		95%		89%		98%		96%		89%	
Humidity	99%		91%		83%		97%		93%		84%	
Weather	98%	a	88%	ab	81%	b	96%	a	92%	ab	80%	b
Daily amount of light	98%	a	85%	a	62%	b	97%	a	90%	a	69%	b
Noise	98%	a	84%	b	63%	c	98%	a	86%	a	63%	b
Smell	97%	a	80%	b	62%	c	94%	a	78%	b	57%	c
Number of respondents *n* =	**279**		**401**		**241**		**279**		**401**		**241**

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
