# Peer review of "Varying Opinions about Animal Welfare in the Australian Live Export Industry: A Survey"

_animals, 2020, doi:10.3390/ani10101864_

Round 1

Reviewer 1 Report

Varying opinions about animal welfare in the Australian live export industry: A survey

The aim of the study is to determinate the perception of stakeholders of animal welfare in order to contribute towards identifying specific animal welfare measures that are perceived as both important and practical considering multiple points of the export chain and different type of livestock.

The subject matter and how it was treated is relevant for the scientific community and to lay new basis in the field of animal welfare.

The manuscript is well written, and the study is well explained and can improve with small changes. The introduction is exhaustive as well as the discussion. Methods and the results are clearly explained. I only have a few comments and suggestions.

Introduction

LL 56 what are the main public concerns related to supply chain and/or LEI?

It is important also to underline that in the last decade, the attention of the consumers on animal welfare is increased in order to make a conscious and ethical choice at the moment to buy an animal product. So, the consumers and the public concerns can influence also the market.

LL 79 delete Duston-Clarke. This reference is the number 19 according with references part

LL 90-91 it should be interesting specify how the perceptions towards animal welfare assessment is not well understood within the industry. This point deserves a better explanation.

Methods

I suggest checking if all the reported results are anticipated in the methods

LL 126 Were the VAS (visual analogue scale) scales also used in this case?

Results

LL 202 scientific journals means popular scientific journals or journals with impact factor and peer-review procedure?

LL 220 animal roles: it is not present in methods. What scale was used? VAS?

LL 230 figure 6 instead of figure 4?

LL 232 Figure 6a, 6b, 6c?

LL 233 Figure 6y and 6z?

LL 268-269 this point (concerns about livestock within LEI) seems not included in methods. I suggest explaining also the scale used for this question.

LL 282-283 the same for this point: the management of data (publicly available or confidential)

Discussion

LL 353-354 I suggest putting some examples of measures

Table 1: the sum of the animal welfare measures in the table would be 35.

LL 414 it could be useful and relevant for the discussion explain better the importance of animal-based measures in the animal welfare assessment

Author Response

Reviewer 1

Varying opinions about animal welfare in the Australian live export industry: A survey

The aim of the study is to determinate the perception of stakeholders of animal welfare in order to contribute towards identifying specific animal welfare measures that are perceived as both important and practical considering multiple points of the export chain and different type of livestock.

The subject matter and how it was treated is relevant for the scientific community and to lay new basis in the field of animal welfare.

The manuscript is well written, and the study is well explained and can improve with small changes.

  • Thank you for the comments

The introduction is exhaustive as well as the discussion. Methods and the results are clearly explained. I only have a few comments and suggestions.

  • Thank you. We feel that the Introduction and Discussion are indeed ‘thorough’, but if there is a specific need to address the ‘exhaustive’ comment (if it’s not made apparent in the following comments/suggestions) then we will attempt to do so.

Introduction

LL 56 what are the main public concerns related to supply chain and/or LEI?

  • We believe that this was addressed in the opening statement of this paragraph: “The general public currently seeks greater assurance about the management and welfare outcomes of exported animals [3].”

It is important also to underline that in the last decade, the attention of the consumers on animal welfare is increased in order to make a conscious and ethical choice at the moment to buy an animal product. So, the consumers and the public concerns can influence also the market.

  • This paragraph deals directly with this issue.

LL 79 delete Duston-Clarke. This reference is the number 19 according with references part

  • Apologies, this was an Endnote error

LL 90-91 it should be interesting specify how the perceptions towards animal welfare assessment is not well understood within the industry. This point deserves a better explanation.

  • We have reworded this sentence to capture that these attitudes and perceptions have not been investigated.

Methods

I suggest checking if all the reported results are anticipated in the methods

  • Thank you for this suggestion. We have double checked, but if there was a specific method that seemed to be missing, we would appreciate this being brought to our attention.

LL 126 Were the VAS (visual analogue scale) scales also used in this case?

  • We have added further details.

Results

LL 202 scientific journals means popular scientific journals or journals with impact factor and peer-review procedure?

  • Not specified, so this could have meant both. We have included a note indicating so.

LL 220 animal roles: it is not present in methods. What scale was used? VAS?

  • Apologies – this has been added.

LL 230 figure 6 instead of figure 4?

  • Great pickup, thanks – fixed.

LL 232 Figure 6a, 6b, 6c?

  • Great pickup, thanks – added.

LL 233 Figure 6y and 6z?

  • Great pickup, thanks – added.

LL 268-269 this point (concerns about livestock within LEI) seems not included in methods. I suggest explaining also the scale used for this question.

  • Added

LL 282-283 the same for this point: the management of data (publicly available or confidential)

  • We have added a bit more detail in the methods to make this clearer.

Discussion

LL 353-354 I suggest putting some examples of measures

  • Great suggestion. We have added “Eg. body temp and heart were seen as only moderately important and not very practical by the LEI group, but much more highly rated by the AWA and GP groups.”

Table 1: the sum of the animal welfare measures in the table would be 35.

  • Corrected, thanks

LL 414 it could be useful and relevant for the discussion explain better the importance of animal-based measures in the animal welfare assessment

  • We have added a comment to this effect, with citation to a relevant study.

Reviewer 2 Report

Varying opinions about animal welfare in Australian live export industry: A survey

This manuscript detail a survey outlining opinions on the Live Export Industry from a variety of different stakeholders. Overall, a very interesting manuscript, great clarity with only a few editing suggestions, as shown below.

Introduction

Line 79 Correct referencing

Results

A general comment on the statistical reporting. Is it possible, where appropriate, to indicate which of the stakeholder groups were significantly different to each other? Some examples below.

Line 146 The majority of the AWA and GP groups were both interested in LEI due to their concern for animals although the statistical figures indicate significant differences. Is this between these two groups and the LEI group or between the AWA and GP groups? If it’s between the AWA and GP groups I’m not following how they can both be interested yet be statistically different? Is there a different criteria/threshold you are using to indicate they are interested if they are statistically different?

Line 179 Are all stakeholder categories significantly different to each other or are just the AWA and LEI significantly different to each other as the GP category is proportionally distributed across urban and rural locations?

Line 183 The manuscript indicates there were no significant difference in education level between the three stakeholder categories although the p-value indicates otherwise?

Line 280 Consider if this figure needs a statically statement similar to that which you have for figure 6 and others.

Author Response

Reviewer 2

This manuscript detail a survey outlining opinions on the Live Export Industry from a variety of different stakeholders. Overall, a very interesting manuscript, great clarity with only a few editing suggestions, as shown below.

  • Thank you very much.

Introduction

Line 79 Correct referencing

  •  

Results

A general comment on the statistical reporting. Is it possible, where appropriate, to indicate which of the stakeholder groups were significantly different to each other? Some examples below.

  • Thanks for raising this – we have attempted greater clarity throughout.

Line 146 The majority of the AWA and GP groups were both interested in LEI due to their concern for animals although the statistical figures indicate significant differences. Is this between these two groups and the LEI group or between the AWA and GP groups? If it’s between the AWA and GP groups I’m not following how they can both be interested yet be statistically different? Is there a different criteria/threshold you are using to indicate they are interested if they are statistically different?

  • We have added that the LEI were not asked this question and have re-worded the sentence.

Line 179 Are all stakeholder categories significantly different to each other or are just the AWA and LEI significantly different to each other as the GP category is proportionally distributed across urban and rural locations?

  • We have reworded and clarified. Thanks for picking this up.

Line 183 The manuscript indicates there were no significant difference in education level between the three stakeholder categories although the p-value indicates otherwise?

  • Thanks for picking that up – the p value had not been correctly transcribed from the statistics.

Line 280 Consider if this figure needs a statically statement similar to that which you have for figure 6 and others.

  • We carried out chi tests for each of the species/location categories – all were p<0.001. We have added a bit more description to make this result more obvious.